



# Multicore structures and the splitting and merging of eddies in global oceans from satellite altimeter data

Wei Cui [1, 2], Wei Wang [1], Jie Zhang [2], and Jungang Yang [2]

[1] Physical Oceanography Lab, Qingdao Collaborative Innovation Center of Marine Science and Technology, Ocean University of China, Qingdao, People's Republic of China
[2] The First Institute of Oceanography, State Oceanic Administration, Qingdao 266061, China

*Correspondence to*: Wei Cui (cuiwei@fio.org.cn)

**Abstract.** This study investigated the statistics of eddy splitting and merging in the global oceans based on 23 years' altimetry data. Multicore structures were identified using an improved a geometric closed-contour algorithm of sea surface height. Splitting and merging events were discerned from continuous time series maps of sea level anomalies. Multicore structures represent an intermediate stage in the process of eddy evolution, similar to the generation of multiple nuclei in a cell as a preparatory phase for cell division. Generally, splitting or merging events can change substantially (by a factor of two or more) the eddy scale, amplitude, and eddy kinetic energy. Specifically, merging (splitting) generally causes an increase (decrease) of eddy properties. Multicore eddies were found to tend to split into two eddies with different intensities. Similarly, eddy merging is not an interaction of two equal-intensity eddies, and it tends to manifest as a strong eddy merging with a weaker one. A hybrid tracking strategy based on the eddy overlap ratio, considering both multicore and single-core eddies, was used to confirm splitting and merging events globally. The census revealed that eddy splitting and merging do not always occur most frequently in eddy-rich regions, e.g., their frequencies of occurrence in the Antarctic Circumpolar Current and western boundary currents were found to be greater than in mid-latitude regions ($20°$–$35°$) north and south. Eddy splitting and merging are caused primarily by an unstable configuration of multicore structures due to obvious current– or eddy–topography interaction, strong current variation, and eddy–mean flow interaction.

## 1 Introduction

Mesoscale eddies are large bodies of swirling water, which generally refer to ocean signals with spatial scales of tens to hundreds of kilometers and temporal scales of days to months (Robinson, 2010). Eddies are found nearly everywhere in the global oceans (Chelton et al., 2011; Cheng et al., 2014; Fu, 2009), and they dominate the ocean's kinetic energy (Morrow and Le Traon, 2012). Following recent advances in remote sensing satellites and the abundance of in situ observational data, it has been established that mesoscale eddies transport water, heat, salt, and energy as they propagate in the oceans (Dong et al., 2014; Thompson et al., 2014; Xu et al., 2011). By combining satellite altimetry and Argo profiling float data, Zhang et al. (2014) found that eddy-induced zonal mass transport was comparable in magnitude to that of the large-scale wind- and thermohaline-driven circulation, which suggested mesoscale eddies have a strong impact on global climate change and air-sea interaction. Mesoscale eddies also have an important influence on the local circulation of the marginal sea, such as the South China Sea (Zheng et al., 2017), the Bay of Bengal (Cui et al., 2016), the Mediterranean Sea (Escudier et al., 2016).



In many previous studies, eddies have been treated as independent water bodies without consideration of eddy–eddy interaction. The study on formation and dissipation of the eddies suggested that dynamic activities of the eddies are mainly due to baroclinic instabilities of the mean flow, topography affects, fluctuating surface winds (Fu et al., 2010; Stammer and Wunsch, 1999). In fact, the eddy dynamics in the ocean are more complicated. Some studies found that an eddy's termination are attributed to many reasons, including frictional decay, eddy-mean interaction and coalescence

with other eddies (Adcock and Marshall, 2000; Morrow et al., 1994; Trieling et al., 2005; Zheng et al., 2011). Schonten et al. (2000) monitored 20 rings which had lifetime greater than 5 months and analyzed their traces using TOPEX/Poseidon altimetry. They found that 13 rings were generated by split off from other rings and three rings split once, one split twice and two even split four times. Fang and Morrow (2003) investigated characteristics of eddies in the Leeuwin Current, they found that interaction with topography can induce splitting or merging of eddies which further

affects the eddy decay. Eddies from different processes may coalesce and form a single eddy due to complicated eddy–eddy interaction (Dritschel and Waugh, 1992; Griffiths and Hopfinger, 1987; Nan et al. 2011). Zhai et al. (2010) modeled a random sea of eddies which propagate westward in the ocean, and the simulation result showed that the eddies interact with one another and cascade energy to larger scales through the merging of eddies of the same parity and finally dissipate near the western boundary.

Studies show that eddy–eddy interaction is universal within the ocean (Trieling et al., 2005; Prants et al., 2011). A very small number of studies have investigated localized eddy splitting and merging, confirming eddy variation through traditional visual interpretation of sea surface height fields (Fang and Morrow, 2003; Schonten et al., 2000). Such studies simply considered an eddy at one moment as a single eddy entity, which was then split into two separate eddies at the next moment, without consideration of eddy–eddy interaction processes. Although such a simplified solution can reveal

the dynamic behavior of eddies, the evolutionary process remains obscure. Some studies of eddy–eddy interaction have found abundant multicore eddy structures within the global oceans (Du et al., 2014; Le Vu et al., 2018; Trieling et al., 2005; Yi et al., 2014a). Generally, multicore structures, which have two or more closed eddies of the same polarity within their boundaries, represent an important transitional stage in which the component eddies might experience splitting, merging, or other energy-transferring interactions. In studying eddy–eddy interaction processes, clear

identification of multicore eddy structures is necessary.

Over the last ten years, researchers have achieved eddy identification automatically from large remote sensing datasets and in situ datasets in many ways (Dong et al., 2011; Liu et al., 1997; Matsuoka et al., 2016; Sadarjoen and Post, 2000). Especially eddy identification and tracking from sea surface height fields has already developed maturity and has been applied to actual eddy studies (Chaigneau et al., 2008; Isern-Fontanet et al., 2003; Nencioli et al., 2010). More and

more researches show a purely geometric method that based on sea level anomaly (SLA) is more accurate and is becoming more of a mainstream method in recent years (Faghmous et al., 2015; Souza et al., 2011; Sun et al., 2017). The purely geometric method for eddy identification is stems from Chelton et al. (2011), and some developments and practical improvements have been made by many researchers (Cui et al., 2016; Mason et al., 2014; Schlax and Chelton, 2016). Chelton et al. (2011) recognized that their original identification algorithm can yield eddies with more than one

local extremum of SLA. They attempted to separate these multiple eddies and only found extra undesirable problems in eddy tracking, so they eventually abandoned the separating procedure. Note that such multiple eddies are very common



in SLA data (Li et al., 2014; Wang et al., 2015; Yi et al., 2014a), this problem can occur when eddies are physically close together. For these multiple eddies, Yi et al. (2015) presented a Gaussian-surface-based approach to identify and characterize the multicore structures of eddies from SLA datasets and results of detecting dual-eddy structures in the

South China Sea demonstrate the effectiveness of the identification approach. But merely identifying the multicore eddy structures is not enough, tracking them based on kinematic properties through time allows us to analyze these multicore eddies as either merging or splitting, how they merge or split and how eddies interact with each other. We believe that revealing the mixing process of eddy splitting and merging in the ocean will have a positive effect on our knowledge about ocean mesoscale dynamic process.

Based on sea level anomaly (SLA) data acquired over a 23-year period (January 1993 to December 2015), this study used a threshold-free closed-contour algorithm to identify mesoscale eddies and multiple eddies within the global oceans. Multiple eddies were confirmed as multicore eddy structures through two-dimensional anisotropic Gaussian surface fitting. Based on the sequential kinematic properties of all eddies, the splitting and merging processes of eddies were analyzed. Moreover, remote sensing sea surface temperature (SST) data were used to validate the eddy–eddy

interactions. The remainder of this paper is organized as follows. Section 2 describes the satellite data used, as well as the methods adopted for eddy detection and multicore eddy confirmation. Section 3 provides two examples of eddy splitting and merging and it describes their evolutionary processes. Section 4 reports global statistics of eddy splitting and merging and highlights the average changes of eddy properties. Finally, a summary and conclusions are presented in Section 5.

**2 Data and Methods**

### 2.1 Altimeter-derived SLA data and AVHRR SST data

The presence and positions of mesoscale eddies were determined by analyzing SLA fields, merged and gridded multimission altimeter products, which are distributed by Archiving Validation and Interpretation of Satellite Data in Oceanography (AVISO). The daily SLA fields with spatial resolution of 0.25 ° over the global oceans spanned the

23-year period from January 1993 to December 2015. Filtering processes were used to remove residual noise and small-scale signals from the AVISO data (Dufau et al., 2013); thus, only the large-scale and mesoscale signals were retained within the SLA fields.

Advanced Very High Resolution Radiometer (AVHRR) SST data were adopted to validate eddies identified using the SLA fields. The AVHRR SST data comprised merged and gridded monomission products using optimal interpolation,

which were provided by the National Oceanic and Atmospheric Administration (NOAA) with the same temporal and spatial scales as the AVISO gridded products. Here, to identify mesoscale variabilities in the ocean, the SST anomaly (SSTA) was constructed by removing the climatological mean and seasonal cycles.



### 2.2 Methods

#### 2.2.1 Eddy identification and some improvements

The SLA fields include a wide range of ocean features, ranging from mesoscale to large-scale ones. The large-scale or low-frequency SLA variabilities were removed from the original SLA data using a high pass filter to produce a grid which includes only mesoscale variability (Chaigneau and Pizarro, 2005; Chelton et al., 2011). A map of SLA comprised of $0.25° \times 0.25°$ pixels is thus obtained at each daily time step. Considering the geostrophic balance, the zonal and meridional surface velocity components $u'$, $v'$ are calculated from the SLA fields, $u' = -\frac{g}{f}\frac{\partial(SLA)}{\partial y}$, $v' = \frac{g}{f}\frac{\partial(SLA)}{\partial x}$, in

which $g$ is the gravitational acceleration, $f$ is the Coriolis parameter, $\partial x$ and $\partial y$ are the eastward and northward unit distances respectively.

Oceanic mesoscale eddies can generally be identified as regions enclosed by SLA contours within which waters of unique characteristics are trapped and subsequently translated. A purely geometric algorithm for eddy identification based on the outermost closed contour of an SLA has been proposed by Chelton et al. (2011). Similar to Chelton et al.

(2011), we defined a closed SLA contour and its internal grid points as an eddy when the following criteria were satisfied:

(1) The SLA values of all of the internal grid points were above (below) that of the outmost closed SLA contour for anticyclonic (cyclonic) eddies.

(2) The number of internal grid points was > 8 and < 1000.

(3) There was at least one and at most three local maximum (minimum) points of SLA for anticyclonic (cyclonic) eddies. The local extremum points were seen as eddy centers.

(4) The amplitude of the eddy was at least 3 cm (Chaigneau et al., 2011; Cui et al., 2016). The amplitude of an eddy was defined as the absolute value of the SLA difference between the eddy center and its edge. For multiple eddies that have more than one center, the amplitude was defined as the maximum SLA difference.

(5) The distance between the two furthest-apart internal points was less than a specified maximum for an eddy. Distance max = 600 km for latitudes below 25°, or 400 km for latitudes above 25° (Schlax and Chelton, 2016).

(6) The eddy edge was defined as the closed SLA contour for which rotational speed $U$ was greater than the translation speed $c$ in the ocean. In the strong Western Boundary Current regions, a value of $c = 10$ cm/s was adopted. In the open ocean, a value of $c = (10 - 7 \times latitude/50)$ cm/s was adopted, which varied with latitude (Fu, 2009; Fu et al.,

2010). For latitudes above 50°, a constant value of $c = 3$ cm/s was adopted (Chelton and Schlax, 1996). Here, the rotational speed $U$ of a closed contour was considered the average of the geostrophic speed of all points in the contour, and the translation speed $c$ referred to the change of eddy position as a function of time.

Note that, the criterion 6 was the largest difference from Chelton et al. (2011). Chaigneau et al. (2011), Flierl (1981) and Yang et al. (2013) suggested that the boundary of eddy water was determined by the ratio of rotational speed to

translation speed, and that an eddy advects the interior water with itself when the ratio is great than 1. Although Chelton et al. (2011) used the outermost contour as the eddy boundary, a comparison of maximum rotational speed $U$ and translation speed $c$ made them believe that the most essential feature of mesoscale eddies is nonlinearity. That means



when this value of $U/c$ exceeds 1, an eddy can advect trapped fluid within its interior as it translates, even the heat, salt and potential vorticity, as well as biogeochemical properties such as nutrients and phytoplankton (Chelton et al., 2011; Samelson, 1992). Therefore, the definition of the eddy boundary as the closed contour with rotational speed $U$ exceeding translation speed $c$, in which the coherent mesoscale features are preserved, is appropriate and accurate.

Besides, for criterion 4 the minimum amplitude of an eddy was increased from the original 1 cm used by Chelton et al. (2011) to 3 cm in this study. The reason for this change was that the accuracy of measuring heights using Jason series altimeters (including Topex/Poseidon and Jason-1/2/3), which currently have optimal performance for observing ocean dynamics, is only about 2 cm in the open sea (Dufau et al., 2016). Therefore, even though the AVISO gridded SLA products represent the merging of data from different altimeters, it is difficult to claim that ocean signals under a variance of 2 cm could be captured precisely in the SLA fields, especially for the gaps in altimeter tracks that are interpolated from other observation points. This change avoided large, ameba-like eddy structures effectively because eddies with amplitude less than 3 cm tended to be broad and relatively flat (Cui et al., 2016; Dufau et al., 2016).

Here, the definitions of some properties for an eddy were given. Similar to Chaigneau et al. (2008), the eddy size is represented by the eddy area $A$, which is delimited by the closed eddy boundary. The eddy scale/radius $R$ corresponds to the equivalent radius of a circle that has the same area as the region within the eddy perimeter, which is $R = \sqrt{A/\pi}$. The eddy intensity or eddy energy can be quantified through the mean eddy kinetic energy EKE, which is EKE $= \frac{1}{2}(u'^2 + v'^2)$.

Criterion 3 requires at least one local extremum point of SLA in an eddy interior. Here, we limited the number of extremum points to 1–3. Experience from experimentation has indicated that eddies with more than 3 extremum points are very unstable and short-lived, i.e., they exhibit distinct transformation of shape within a few days. Eddies with 2–3 extremum points generally reflect the period of two or more eddies mixing. Multiple eddies can merge into a single entity through eddy–eddy interactions or a single eddy can split into two separate eddies under the influence of external shear or strain. The identification of multiple eddies is an essential step in studying the splitting and merging processes of eddies. Here, we needed only to identify eddies with one center, which were saved as the single-core eddies in the dataset. Multiple eddies with 2–3 extremum points required further processing to confirm them as a multicore eddy or as single-core eddies in close proximity.

### 2.2.2 Identification of multicore eddy structures

The method of eddy identification in Section 2.2.1 can yield eddies with 2–3 local extremum points of SLA. Multiple eddies could correspond to multicore eddy structures formed because of eddy–eddy interaction and substantial interior-water exchange; however, they could also represent the misidentification of two single-core eddies because of their close spatial proximity and irregularity of the SLA contours associated with noise in the SLA fields. In cases of two single-core eddies without substantial interaction or water exchange, the process for considering a multicore eddy is not reasonable or appropriate. Therefore, a multicore eddy structure must be identified from multiple eddies based on a certain distinguishing method.

The structure of an idealized mesoscale eddy on the sea surface can be considered as a mathematic Gaussian shape,



which is a valid and common method for studying eddy properties and dynamics (Chelton et al., 2011; Maltrud and McClean, 2005; Wang et al., 2015). In this study, a Gaussian surface was adopted for fitting the multiple eddy structures to determine if they constituted real multicore eddy structures or should be considered separate entities. For a multiple eddy structure $H$ with 2–3 extremum points, each component eddy $G_i(x,y)$ that corresponds to one extremum point was fitted by an anisotropic two-dimensional Gaussian kernel expressed as (Yi et al., 2014b and 2015):

$$H = B + \sum_{i=1}^{n} G_i(x, y) , \tag{1}$$

$$G_i(x, y) = A_i \exp\left( -\left( \frac{(x - x_i)^2}{2\sigma_{ix}^2} + \frac{(y - y_i)^2}{2\sigma_{iy}^2} \right) \right) \tag{2}$$

In which, $B$ represents the basal SLA of the multiple eddy structure which equals the SLA value at eddy boundary, $x_i$ and $y_i$ are the positions of the extremum point, $\sigma_{ix}$ and $\sigma_{iy}$ represent the eddy scale along the horizontal and vertical axis, $A_i$ represents the eddy amplitude defined by the difference between the SLA of extremum point and the SLA of boundary, respectively. For a multiple eddy, $B$, $A_i$, $\sigma_{ix}$ and $\sigma_{iy}$ are estimated using the least-square method based on the eddy boundary and all of the eddy interior points. An example case is shown in Figure 1.

For ideal Gaussian eddies, a certain eddy scale $\sigma'$ can be defined as the SLA contour at which the average rotational speed increases to a maximum and the relative vorticity reduces to zero (Chelton et al., 2011; Flierl, 1981; Haller, 2005). If two eddies have significant eddy–eddy interaction and substantial water exchange, their eddy scales $\sigma'$ will have some overlap (Dritschel and Waugh, 1992; Yi et al., 2015). Therefore, the criterion for determining a real multicore eddy structure is that the distance between the composite eddy centers $L(e_1,e_2)$ should be less than the sum of the Gaussian fitting scales ($\sigma_1' + \sigma_2'$) for each eddy pair, which can be expressed as $L(e_1,e_2) < \sigma_1' + \sigma_2'$ ( Yi et al., 2014b and 2015). If a composite eddy fitted by a Gaussian function satisfies the criterion, it can be considered a real multicore eddy (Figure 1). Otherwise the composite eddy structure should be considered as two single eddies because of the misidentification caused by the irregularity of SLA contours, and the identification procedure should proceed upward (downward) for anticyclone (cyclone) with an increment of 1 cm described in Section 2.2.1. Yi et al. (2015) tested the application of the multi-eddy detection algorithm on a series of SLA maps in the South China Sea from 1993 to 2012. The study demonstrated the potential value of the algorithm in helping scientists to investigate characteristics of eddy–eddy interactions from satellite observations.

Application of the eddy identification procedure and the determination method for multicore eddy structures typically detects about 2300 single-core eddies and 200 multicore eddies on average in an arbitrary SLA field globally. The number of all eddies which is about 2500 is a little less than the number of 3000 in Chelton et al. (2011). Considering the use of criteria 4 and 6 in Section 2.2.1, the number is reasonable and acceptable.




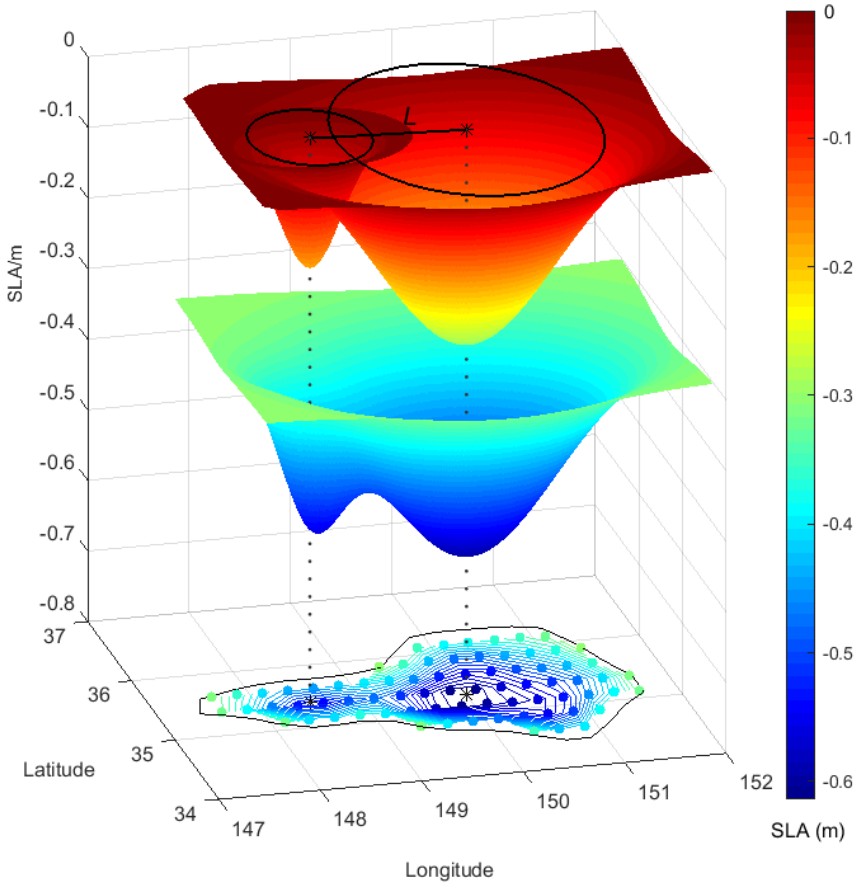

**Figure 1.** Example of fitting a multicore eddy using an anisotropic two-dimensional Gaussian kernel in the SLA field. At the bottom, the
black line represents the multicore eddy boundary, black asterisks represent the eddy centers, dots with color represent the SLA gridded
points within the eddy interior (the colors reflect the value of the SLA), and lines with color represent the SLA contours with 2-cm intervals.
The upper two independent Gauss surfaces $G_1(x,y)$ and $G_2(x,y)$ are fitted using the SLA gridded points, and the middle composite surface is
the superposition of $G_1(x,y)$ and $G_2(x,y)$ with a basal SLA $B$ (here $B$ = -0.3m). The fitting eddy scales $\sigma'$ are shown in black circles at the top,
and $L$ is the distance between the composite eddy centers.

### 2.2.3 Eddy tracking

Many sophisticated algorithms for eddy automated tracking have been widely applied to determine eddy trajectory
(Chaigneau et al., 2008; Henson and Thomas, 2008). In this study, we used the same procedure as Chaigneau et al. (2008)
to track a single-core eddy. For an eddy in the initial map at time $t_0$ day, we search for all eddies in the next map at time $t_0$
+ 1 day and consider the most similar eddy within a spatial radius $R$ as its succession. If there is no eddy matched at $t_0$+1
day, the searching will expend to next time $t_0$ + 2 day, $t_0$ + 3 day, …, until the $t_0$ + 10 day. If the sequential eddy still
cannot be found in 10 days, the eddy at $t_0$ will be considered as the end of the trajectory. The reason of the searching
within 10 days is that sometimes eddies weaken or disappear between consecutive maps if they pass into the gaps
between satellite ground tracks and they may re-emerge in the next couple of days. Considering that the gridded SLA
fields provided by AVISO are merged using three altimeters (most times), the gap between satellite ground tracks is less





than 50 km at 10 °latitude. If an eddy move westward with a speed of 5 cm/s in the ocean, then after 10 days it will have ~40km displacement which is enough to capture the eddy signal at the neighboring ground track.

Generally, oceanic mesoscale processes change slowly. Mesoscale eddies can exist in the global oceans for several months or even years (Chelton et al., 2011). Thus, for multicore eddy structures identified using the process in Section 2.2.2, it is important to examine their stability, i.e., their lifetimes. Based on fluid mechanics, a multicore structure does

not represent a steady state and it will change obviously over time (Overman and Zabusky, 1982; Melander et al., 1988; Dritschel, 1995). Unsurprisingly, the lifetimes of multicore eddies are expected to be shorter than single-core eddies; however, a transient multicore structure that exists in the ocean for only a few days cannot be considered necessarily as a mesoscale process. Based on experience, in this study, only multicore eddy structures that existed for more than 6 days within a 10-day window were considered real eddy structures. In other cases, the multicore structures were considered

transient turbulence signatures within the ocean and they were neglected. Without this step, there could have been a problem with the hybrid eddy tracking including both single-core and multicore eddies. For example, in the early and latter stages of some eddy trajectories, the eddies could exhibit a single-core structure, whereas in the central few days (e.g., 3–4 days), the structure could be multicore. It is difficult to determine that multicore structures could have really formed in the ocean for just a few days; instead, they are more likely to correspond to misidentification of multicore

eddies. In fact, here, we faced an awkward situation; a multicore eddy often has a short lifetime because of its instability, while a multicore structure persisting for just a few days cannot be seen as a real multicore eddy, i.e., multicore structures with longer lifetimes should be considered. This is why we compromised in the multicore eddy tracking. In this study, multicore eddy tracking was conducted before single-core eddy tracking and hybrid eddy tracking (discussed in the next paragraph). Thus, the dataset of multicore eddy trajectories was obtained first.

Hybrid eddy tracking refers to single-core eddies that are independent eddies in the daily results (not trajectories) and multicore eddy trajectories. To match a single-core eddy and a multicore eddy (trajectory), the tracking procedure is different and more complicated compared with that involving just single-core or multicore eddies. The similarity principle is inappropriate considering the significant differences in the properties between single-core and multicore eddies, e.g., their scale or amplitude (Fang and Morrow, 2003). In this study, the spatial attributes of the two types of

eddy were considered for the hybrid tracking, which is similar to the neighbor enclosed area tracking algorithm used for tracking tropical cyclones in the upper troposphere (Inatsu and Amada, 2013). If at time $t_0$ a single-core eddy merges with another to form a multicore structure at time $t_1$, spatial overlap of the multicore eddy and the single-core eddy will be apparent. We used an overlapping ratio $r$, which is the overlapping area of the two eddies divided by the smaller area of the two eddies (generally, this refers to the single eddy), to confirm the evolutionary relationship of a single-core and a

multicore eddy. If the overlapping ratio $r$ of the two eddies is greater than a threshold value $r_0$, the two eddies will be considered as one trajectory. The threshold value $r_0$ varies linearly with the time interval of the next tracked eddy from a maximum of 0.5 (1-day interval) to a minimum of 0.2 (10-day interval) considering both the eddy movement and the identification error of an eddy. The tracking procedure is also applicable to a multicore eddy that splits into two or more single-core eddies.

It should be reemphasized that the multicore eddy trajectories were confirmed first. In fact, in searching for the subsequent stage of a single-core eddy, multicore eddy trajectories were searched first using the hybrid tracking



procedure; if no counterparts were found, then single-core eddies were searched. This way, the hybrid eddy dataset (not the eddy trajectory dataset over the entire lifetime, discussed in Section 4) that includes both single-core and multicore eddies is generated. Moreover, trajectories that included only single-core eddies were saved in the purely single-core

eddy trajectory dataset. A flowchart of the eddy tracking process is shown in Figure 2.

We defined the event of merging of two eddies as follows. If there were two single-core eddies at time $t_0$ (maybe not the same day but within 10 days) matched with a multicore eddy trajectory at time $t_1$ (here, time $t_1$ refers to the start time of the trajectory), the two single-core eddies were considered to merge and form a multicore structure. Similarly, a splitting event was defined when two single-core eddies at time $t_2$ matched a multicore eddy trajectory at time $t_1$ (here,

time $t_1$ refers to the end time of the trajectory). Similar to the discussion regarding the lifetimes of multicore eddies, a multicore eddy trajectory might split into two single-core eddies that persist for only a few days. Such transient single-core eddies are unstable ocean signals that cannot be used in the analysis of the merging or splitting of eddies. Therefore, a multicore eddy trajectory was considered to disappear (or appear) at time $t_1$ and to reflect a turbulence/eddy-like signature in the ocean (not a splitting or merging) if the following (or preceding) single-core eddies

had lifetimes of less than 6 days.

Limited by existing observational data and vortex mixing theory, only eddies with the same polarity were considered in the splitting and merging processes, i.e., we analyzed only the splitting and merging of cyclones, or the splitting and merging of anticyclones. Although a cyclonic eddy could theoretically merge directly with an anticyclonic eddy, the mixing process is too complex and the observation of such an event too difficult for the current research.


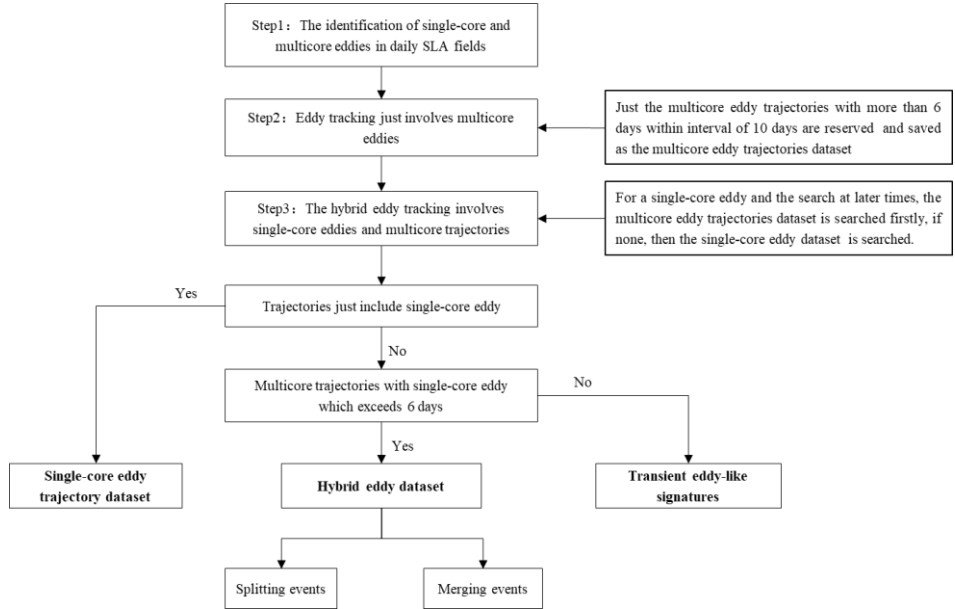

**Figure 2**. Flowchart of the eddy tracking process.



## 3 Cases of Eddy Splitting and Merging and their Evolutionary Processes

Some previous studies of eddy–eddy interaction have found that multicore eddies exist universally within the global

oceans (Li and Sun, 2015; Trieling et al., 2005; Yi et al., 2015). Generally, multicore structures, which have two closed eddies of the same polarity within their boundaries, represent an important transitional stage in their lives during which the component eddies might experience splitting or merging. To elucidate such events and to understand their dynamic processes visually, examples of splitting and merging are examined in this section.

### 3.1 Case of an eddy splitting

The SLA maps of a case of eddy splitting from 1 February to 17 March 2009 are shown in Figure 3. At the start time (1 February), a strong cyclonic eddy existed as a single core in the center of the SLA map. This cyclonic eddy evolved into a multicore eddy because of the southward movement of a strong anticyclone to the north. Through interaction of the two strong eddies, the cyclonic eddy became deformed. The multicore eddy structure persisted for nearly a month before finally splitting into two single-core eddies on 28 February. The two single-core eddies had

smaller scales compared with the multicore eddy before the split, although they gradually became larger as they moved away from each other. Daughter eddy A was larger and it carried more eddy energy, i.e., about 15% of that of the mother eddy, while the smaller daughter eddy B carried only about 5% of the energy. The strong anticyclonic eddy that moved southwestward played an important role in the process of splitting the cyclonic eddy.

Generally, a cyclonic (anticyclonic) eddy corresponds to a cold (warm) core in the SST signature due to upwelling

(downwelling) of centric water (Kubryakov et al., 2018). Consequently, the splitting case based on the SLA maps can be validated through the corresponding SST anomaly (SSTA) maps derived from AVHRR data, as shown in Figure 4. The climatic SST signal has been removed such that the SSTA data can be used to study the oceanic variation corresponding to mesoscale processes. Hence, the local temperature gradients reveal the presence of coherent mesoscale structures. The sequential SSTA maps exhibit signatures similar to the surface oceanic structures in the SLA maps. The four SSTA

snapshots of the upper panels in Figure 4 show that a strong cyclonic (cold core) eddy evolved into a multicore eddy. The four snapshots of the middle panels show this eddy was squeezed and deformed by an anticyclonic eddy, before finally splitting into two eddies. The four snapshots of the lower panels show the evolutions of the two single-core eddies as they moved apart. Thus, the dynamic process is consistent with the SLA results. Note that unlike standard altimetry products, the SSTA field includes many small-scale oceanic signals. It is difficult to remove the unconnected variations

in the SST data, especially diurnal variations that tend to appear as random noise. Diurnal variations of SST of 1 ℃ (and occasionally more) are common in the oceans (Talley et al., 2011). Therefore, even though the SSTA maps display many unstable and unsmooth signals, the strong mesoscale oceanic signals remain apparent.





**Figure 3.** SLA maps of eddy splitting from 1 February to 17 March 2009. Color shading represents the value of the SLA field; arrows represent the surface geostrophic velocity components calculated from the SLA; blue and red lines represent boundaries of cyclonic and anticyclonic eddies, respectively; blue and red dots represent cyclonic and anticyclonic eddy cores, respectively; and black lines represent multicore eddies. For better observing the evolution process, the time interval between two adjacent maps is not the same, in the first row it is 5 days, the last map is after 6 days, and the other intervals are 3 days. The multicore eddy structure persisted for nearly a month before splitting.





**Figure 4.** SSTA maps of eddy splitting 1 February to 17 March 2009 (corresponding to Figure 3). Color shading represents the value of the SSTA field.

The variations of eddy properties for the total splitting process are shown in Figure 5. From the appearance of the multicore eddy structure, the distance between the two cores increased almost monotonically from 80 km at the beginning to almost 200 km at the end (Figure 5a). When the distance between the two cores was > 200 km (day 28), the multicore structure could not restrict the two water masses in its interior; thus, it split into two single-core eddies. The variation of eddy radius (Figure 5b) shows that the scale of the multicore eddy gradually increased as the distance between the cores extended because of tensile deformation of the multicore structure. We found the amplitude (Figure 5c) exhibited slight decline during the latter stages of the multicore structure, which indicated that although the eddy scale increased, the eddy intensity might have weakened. The eddy kinetic energy (EKE) fluctuation of the multicore structure (Figure 5d) implies the two cores were allocated eddy energy in some way. The EKE fluctuation can be explained reasonably through the variations of eddy scale and amplitude. The multicore eddy achieved a balance between increasing scale and weakened amplitude, which resulted in no substantial change of the EKE before the split.

Once the multicore eddy had split into two single-core eddies, the eddy properties changed considerably. The eddy scale or radius decreased substantially from 180 km for the multicore eddy to about 100 km for one eddy and about 50

km for the other, as is also evident in the SLA maps (Figure 3). Although it is difficult to estimate the energy and water exchange between the two daughter eddies, continuing to consider them a superimposed multicore structure is

inappropriate based on the discriminating principle discussed in Section 2.2.2. The two daughter eddies moved away from about 200 km apart to over 300 km apart, and they gradually evolved into two independent stable eddies without interaction (lower panels of Figure 3). The significant reduction of eddy scale after the split caused corresponding reductions in both eddy amplitude and EKE (Figure 5c and d). The amplitude decreased from about 50 cm for the multicore eddy to 22 cm for daughter eddy A and about 12 cm for daughter eddy B. Concurrently, the EKE decreased

from about $2.8 \times 10^5$ cm$^2$/s$^2$ to about $4.2 \times 10^4$ and $1.5 \times 10^4$ cm$^2$/s$^2$ for daughter eddy A and daughter eddy B, respectively.

The substantial variations of eddy properties were closely related to the spatial changes in the shape of the eddy. When the multicore eddy split, the spatial scales of the two daughter eddies instantaneously became much smaller, meaning they captured only some of the original signal and energy of the multicore eddy (Saito and Ueda, 2004). In fact,

most of signal and energy remained "hidden" in the background field. With the evolution of the two single-core eddies, the hidden signal and energy became captured by the two eddies, as evidenced excellently by the increasing scale, amplitude, and EKE of the two independent eddies. It is also suggested that the two single-core eddies gradually evolved into strong stable eddies that were close to the intensity of the original multicore eddy; in fact, daughter eddy A eventually attained a greater amplitude. The four panels of Figure 5 indicate that the evolution of eddy properties from a

single-core eddy to a multicore eddy is smooth and continuous, while that of the splitting process from a multicore eddy to two daughter eddies is discontinuous and irregular.

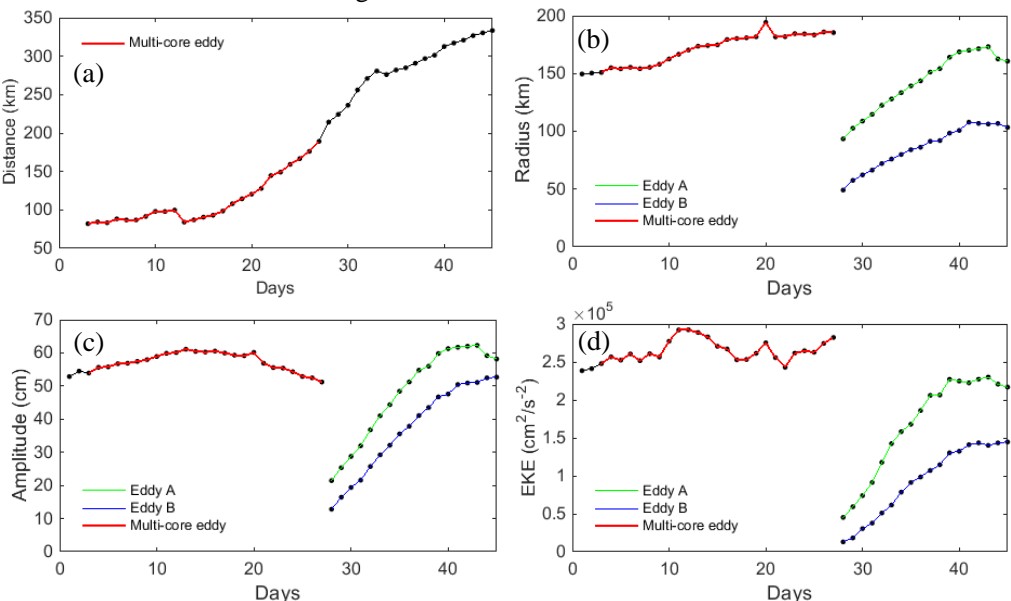

**Figure 5.** Changes of eddy properties with eddy split from 1 February to 17 March 2009: (a) distance between the two cores or two single
eddies, (b) eddy scale, (c) eddy amplitude, and (d) eddy EKE. X axis represents the number of days from 1 February. The multicore eddy existed for 25 days from day 3 (3 February) to day 27 (27 February) and it split into two eddies on day 28 (28 February).





## 3.2 Case of two eddies merging

The SLA maps of two anticyclonic eddies merging from 20 January to 19 February 2012 are shown in Figure 6.
Many numerical studies have recognized three stages in the merging of two ideal vortices: they first rotate around each
other and deform elliptically, then they establish a common boundary and develop a band of vorticity exchange, before
finally merging into a single entity (Huang, 2005; Masina and Pinardi, 1993). The merging process of the two
independent eddies (i.e., eddy A and eddy B) shown in the sequential time series of the SLA maps (Figure 6) is
consistent with these three stages. First, the two eddies approached close enough to each other to instigate obvious eddy–
eddy interaction and cause reduction of their spatial scales. Subsequently, eddy A and eddy B merged into a dual-core
structure that existed for 10 days during 5–14 February. Finally, the dual-core eddy, which was smoothed by diffusion,
evolved into a strong stable single-core eddy. The corresponding SSTA maps derived from AVHRR data are presented in
Figure 7 as validation of eddy merging. The consecutive SSTA maps exhibit eddy signatures similar to the SLA maps. It
should be noted that diurnal variation of the SST signal, as mentioned in Section 3.1, is also evident in Figure 7, which
slightly obscures the mesoscale signals.


**Figure 6.** Same as Figure 3 but showing two eddies merging from 20 January to 19 February 2012. A merged multicore eddy existed for
about 10 days during 5–14 February.



**Figure 7.** Same as Figure 4 but showing two eddies merging, corresponding to Figure 6 from 20 January to 19 February 2012.

The variations of eddy properties for the entire merging process are shown in Figure 8. The distance between the two eddies decreased almost linearly from 360 to 160 km (Figure 8a). When the distance was <160 km, the two eddies merged into a dual-core structure. With the evolution of the dual-core eddy, the distance between the two cores decreased further to <100 km. Then, the dual-core eddy evolved naturally into a single-core eddy on 15 February (day 27; Figure 8). The variation of eddy radius (Figure 8b) showed that the scales of the two separate anticyclonic eddies decreased gradually with the reduction of eddy distance. When the two eddies merged into a multicore eddy, the eddy scale increased from 50 to 110 km and it subsequently fluctuated slightly depending on the changes of shape of the multicore structure. After 10 days, the multicore eddy evolved into a single-core eddy that changed scale smoothly. The variations of eddy amplitude and EKE (Figure 8c and d) showed similar evolutions as the eddy radius. Before merging, the reductions of amplitude and EKE implied the transfer of eddy signal and energy that could not be captured by the two eddies into the background field. However, once the two eddies merged, the multicore eddy recaptured the hidden signal and energy, resulting in the increases of amplitude and EKE. The merging process appears the reverse of the splitting process, i.e., the changes in eddy properties caused by the two eddies merging into a multicore eddy are irregular,



whereas the evolution from a multicore eddy to a single-core eddy is smooth and continuous.

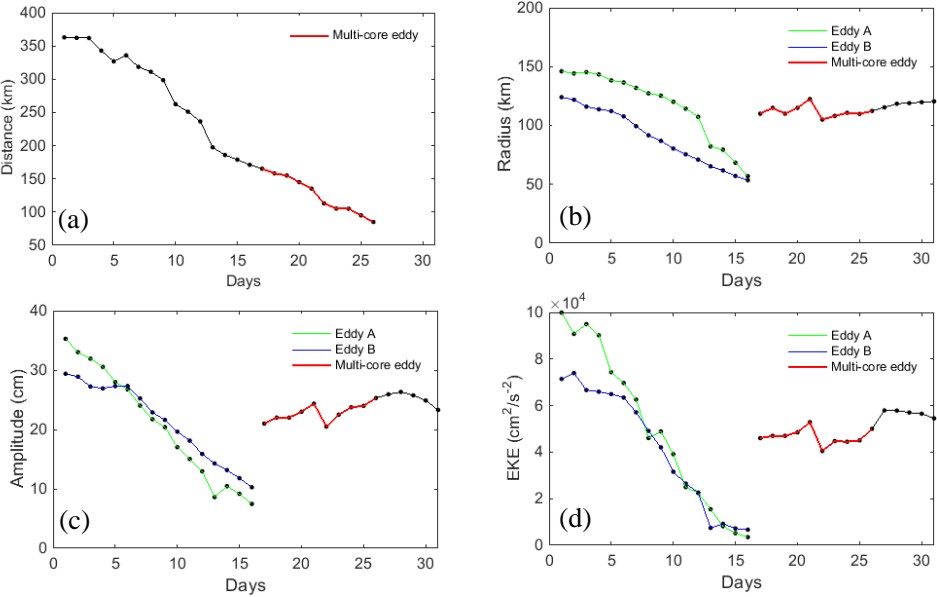

**Figure 8.** Changes of eddy properties with merging of two eddies from 20 January to 19 February 2012: (a) distance between the two cores or two single eddies, (b) eddy scale, (c) eddy amplitude, and (d) eddy EKE. X axis represents the number of days from 20 January. The multicore eddy existed for 10 days from day 17 (5 February) to day 26 (14 February) and evolved into a single-core eddy at day 27 (15 February).

Generally, splitting or merging events can change substantially (by a factor of two or more) eddy scale, amplitude and energy EKE. In another words, the obvious variation of eddy properties for one full-lifetime eddy could correspond to a splitting or merging event (certainly, maybe also to other events, e.g., eddy-current interaction and topographic influence). Eddy–eddy interactions in the oceans are very complex. For example, merging is not limited to two eddies and it could include interactions between three or more eddies merging into a single entity (Saito and Ueda, 2004; Zhai et al., 2010). Moreover, splitting and merging might occur at the same time, e.g., two or more eddies might interact and merge into a multicore structure before the multicore eddy subsequently splits into two or more eddies. Despite these possibilities, the limitations of the eddy identification method and eddy kinetic theory make it very difficult to study such complicated eddy–eddy interactions. Thus, this study focused only on the classical merging of two eddies into one and splitting of one eddy into two, which are perhaps the most representative eddy–eddy interactions in the oceans. Multicore structures are vital intermediate stages in the processes of eddy splitting and merging. The next section discusses the execution of hybrid tracking of single-core and multicore eddies over the 23-year period from January 1993 to December 2015, and it examines the census of splitting and merging events globally.

# 4 Global Statistics of Eddy Splitting and Merging

Statistical analysis of all multicore eddy trajectories identified by the automated tracking procedure, without hybrid tracking, over the 23-year period (January 1993 to December 2015) revealed 83,751 cyclonic and 83,406 anticyclonic multicore eddies with lifetimes >6 days. While there are ~ 250,000 eddies with lifetimes >30 days and ~ 40,000 eddies with lifetimes >100 days for single-core eddies over the 23-year period. About 95% of the multicore eddies had lifetimes



of <30 days, and about 97% of the multicore eddies had propagation distances of <200 km. In comparison with single-core eddies (AVISO, 2017; Chelton et al., 2011), the lifetimes and propagation distances of multicore eddies tend
to be much smaller. These multicore structures are more likely to interact with background fluids and they are easily affected by other eddies in the ocean. Therefore, such structures are easily deformed and they tend to split into two eddies or merge into one eddy, or even dissipate directly into the background fields.

        Hybrid tracking is a complicated process when considering both single-core and multicore eddies throughout their full evolutionary lifetimes. If an eddy splits or merges just once during its lifetime, its evolutionary process could be
easily discerned. However, some eddies might merge or split many times during their lifetimes. Schonten et al. (2000) tracked 20 eddy rings with lifetime exceeding 5 months in the Agulhas retroflection and found three of the original 20 split once, one split twice and two even split four times. Fang and Morrow (2003) studied the evolution and decay of 37 eddies originating in the Leeuwin Current and found one eddy split into two eddies, one of which in turn split into two new eddies. Above-mentioned tracking only is limited to a finite number of eddies (20 and 37 respectively) that is easy
for full-lifetime eddy tracking. Full lifetime global eddy hybrid tracking involves millions of eddy trajectories and it is almost impossible for current research. Eddy splitting and merging multiple times can cause an increase in complexity of one order of magnitude. It is not possible to have an effective global means for describing the evolutions of eddies that might merge or split multiple times during their full lifetimes. To investigate splitting and merging events, this study considered a certain time (at least 10 days) before and after the presence of multicore structures was determined, so that
they could be observed clearly in the evolutionary process.

        In total, the eddy hybrid tracking procedure, summarized in Section 2.2.3, detected 47,312 splitting events and 50,166 merging events for all 167,157 multicore eddies (Table 1). Specifically, there were 24,008 cyclonic and 23,304 anticyclonic multicore eddies that split into two eddies (fewer than 10 split into three eddies), which accounted for 28.3% of the total. Similarly, there were 25,709 cyclonic and 24,457 anticyclonic multicore eddies that merged into one eddy,
which accounted for 30.0% of the total, i.e., slightly more than the number of splitting events in the global oceans.

        It is important to note that there were 46,936 multicore eddies identified as part of the evolution of single-core eddies that did not split or merge (which means the eddies before and after the multicore structures were all single core). These multicore eddies represent intermediate states of single-core eddy evolution. They tend to have shorter lifetimes and greater SLA differences between the two cores, which cause the stronger core to absorb the weaker core directly.
Moreover, the change in eddy properties from a single-core eddy to a multicore eddy (or the reverse) is smooth and continuous; there is no abrupt change in properties as in the splitting and merging events discussed in Section 3. Furthermore, there were 22,743 multicore eddies with transient eddy-like signatures (Section 2.2.3). These 22,743 multicore eddies had an average amplitude of about 5 cm, while that of the multicore eddies involved in splitting and merging events was about 16 cm, indicating considerable difference in eddy intensity. Therefore, these weaker eddies
with an average radius of about 115 km exhibited a large-scale relatively flat pattern, and they tended to disappear directly under interaction with the background field.



**Table 1.** The classification of all multicore eddies (exactly, trajectories) based on hybrid tracking and their numbers

| Multicore eddy type | Splitting | Merging | transient eddy-like signatures | Part of single-core eddies | All |
|---|---|---|---|---|---|
| Cyclonic | 24008 | 25709 | 11611 | 22423 | 83751 |
| Anticyclonic | 23304 | 24457 | 11132 | 24513 | 83406 |
| Total | 47312 28.3% of all | 50166 30.0% of all | 22743, 13.6% of all | 46936 28.1% of all | 167157 |

This study focused on multicore eddies that experienced splitting or merging. Geographic frequency statistics of the 47,312 splitting events and 50,166 merging events are shown in Figure 9. The upper and lower panels show the numbers of splitting and merging events, respectively, which occurred in $1\degree \times 1\degree$ regions (smoothed using a $3\degree \times 3\degree$ window) during the 23-year study period. The geographical patterns of the census of merging and splitting events are very similar. Globally, splitting and merging tend to occur in the Antarctic Circumpolar Current and western boundary currents (refer

to the Gulf Stream and its extension, and in the region of confluence of the Kuroshio and Oyashio Currents and their eastward extensions, the Agulhas Return Current, and the Brazil–Malvinas Confluence), where typically about 5–7 events are observed. Specifically, for the Northern Hemisphere, splitting and merging mainly occur in the Gulf Stream and its extension, the Kuroshio Extension, the region of Subtropical Countercurrent in the Northwest-Pacific, the eastern Pacific coastal region, and the region in the Gulf of Mexico. For the Southern Hemisphere, splitting and merging mainly

occur in two zonal bands. One is the region between 20°S and 35°S because of obvious eddy–mean flow and eddy–eddy interaction related to the South Equatorial Current variations (Qiu and Chen, 2004), in which new-formed eddies or multicore eddies prefer to propagate westward with movement of the background current. Another zonal band is the Antarctic Circumpolar Current region, which is affected by the strong eastward Antarctic Circumpolar Current and the significant air-sea interaction; eddy–eddy interaction is more frequent here (Frenger et al., 2015). Less splitting and

merging in two zonal band, about 35°– 45°S in the southern Indian Ocean and 40°– 50°S in the southern Pacific Ocean, is evident in Figure 9. The low frequency distribution of splitting and merging is easy to understand because oceans there are usually calm without forcing mechanism to cause eddy–eddy interaction. Unsurprisingly, few splitting and merging events occur throughout the equatorial region because few eddies are observed in that region. Another noteworthy feature is the higher numbers of splitting and merging events (up to about 10) that occur in the Drake Passage, to the

west of the Kerguelen Plateau, to the south of New Zealand, in the Gulf of Mexico, and along the axis of the Gulf Stream. The higher numbers in these regions are closely related to eddy–topography or current–topography interactions (Adcock and Marshall, 2000; Frenger et al., 2015), especially in the area of the dramatically narrowed Drake Passage across which eddies propagate eastward.




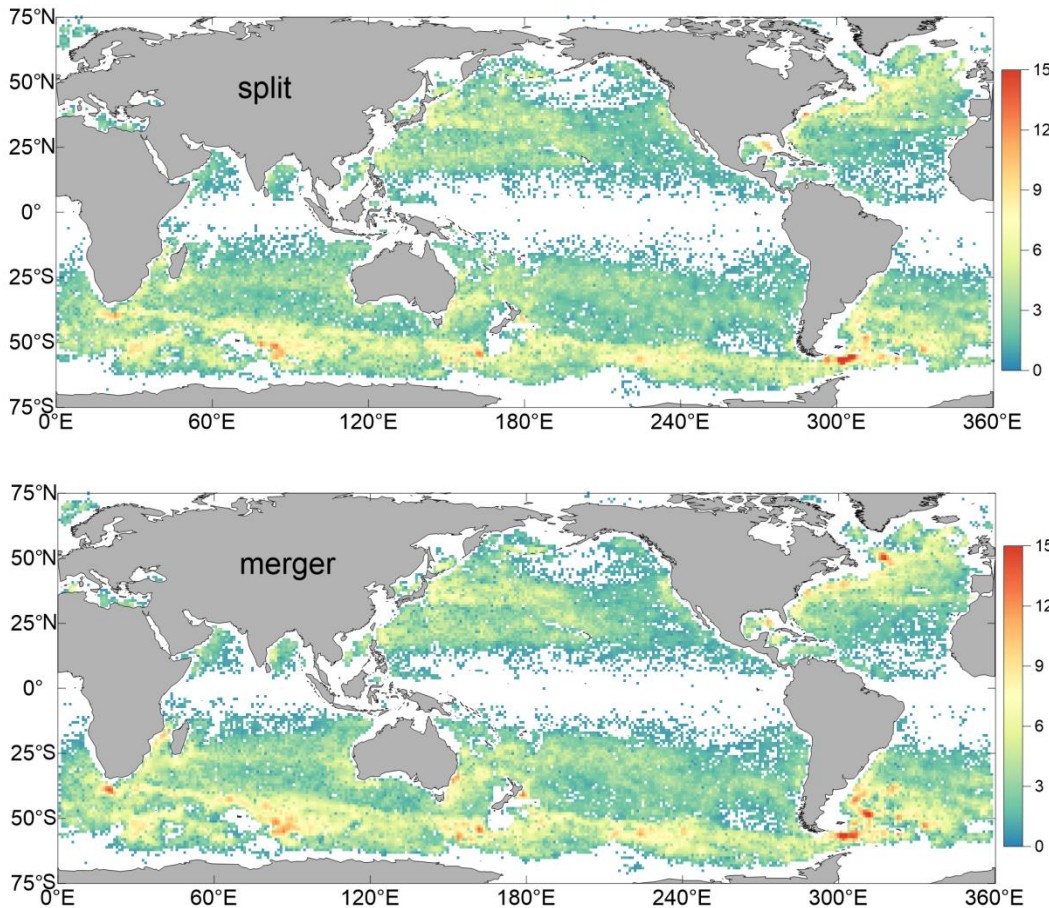

**Figure 9.** Census statistics for eddy splitting and merging events for each $1\,° \times 1\,°$ region (smoothed using a $3\,° \times 3\,°$ window) over the 23-year period (January 1993 to December 2015).

Eddy splitting and merging events do not always occur most frequently in eddy-rich regions. Compared with the geographical distribution of global eddies (Figure 5 in Chelton et al., 2011), the mid-latitude regions of $20\,°$–$35\,°$ north and south have significantly higher eddy frequency, but not the highest frequencies of eddy splitting and merging. Eddies in the mid-latitudes tend to have long lifetimes, which implies the ocean currents and eddy structures are stable with less (not zero) variation. Consequently, eddy–eddy interaction and eddy splitting and merging are not as significant as eddy frequency. In addition, the Antarctic Circumpolar Current, the Gulf Stream and its extension, Kuroshio Extension, Agulhas Return Current, and Brazil–Malvinas Confluence Zone have relatively fewer long-lifetime eddies but higher frequencies of splitting and merging than mid-latitude regions. This implies that regions with strong current variation and obvious eddy–mean flow interaction where abundant eddies have shorter lifetimes, have more significant eddy-eddy interaction and more splitting and merging events. These interactions are more likely to cause unstable configuration of an eddy (e.g., multicore eddy) and eddy–eddy interaction, and then the eddy merging or splitting occur (Griffiths and Hopfinger, 1987; Trieling et al., 2005). In turn, eddy lifetimes based on previous eddy tracking without consideration of eddy–eddy interactions are likely to be shorter than their real lifetimes. For example, the split of a single eddy into two eddies would be considered the end of the eddy under the previous method, which is obviously inappropriate.



The variations of eddy properties for all splitting and merging events are shown in Figure 10. The properties of the single-core eddies were normalized with respect to the properties of the multicore eddies. For eddy splitting, there is slight increase in terms of the properties of the multicore eddy stage, probably because the two cores stretch the

multicore structure to store energy for eddy splitting. Once the multicore eddy has split into two single-core eddies, the eddy properties are reduced considerably. The radius of each of the two daughter eddies is half that (or even smaller) of the mother eddy and between them, the eddy amplitude and eddy kinetic energy differ greatly. The amplitude of the larger daughter eddy is almost twice that of the smaller one; moreover, they contain about 30% and 10% of the original eddy energy, respectively, and the remaining energy is transferred to the background field. With the evolution of the two

single-core eddies, the hidden signal and energy became captured by the two eddies, as evidenced excellently by the increasing scale, amplitude, and EKE of the two independent eddies. It is also suggested that the two single-core eddies gradually evolved into strong and stable eddies.

For eddy merging, two single-core eddies with similar radii but different intensities gradually decrease in terms of radius, amplitude, and EKE as the intervening distance decreases. It shows that the two eddies interact and that some of

their energy is hidden in the background field, which causes a reduction in the eddy properties, especially the EKE. When the two eddies merge into a multicore eddy structure, the eddy properties change substantially. The eddy radius is almost doubled, indicating that the eddy area could increase by 3–4 times. The multicore eddy recaptured the hidden signal and energy from background field, resulting in the increases of amplitude and EKE. The two single-core eddies do not differ greatly in spatial scale, but one eddy is much larger than the other in terms of eddy intensity (amplitude and

EKE), i.e., the large one is nearly double the smaller one. It shows that eddy merging is not an interaction of two equal-intensity eddies, and that it tends to manifest as a strong eddy merging with a weaker one to form a larger multicore eddy in a process that appears the reverse of splitting. Generally, splitting or merging events can change the eddy scale, amplitude, and EKE substantially. In other words, the obvious variation (twice or more) of eddy properties for one full-lifetime eddy could correspond to a splitting or merging event.

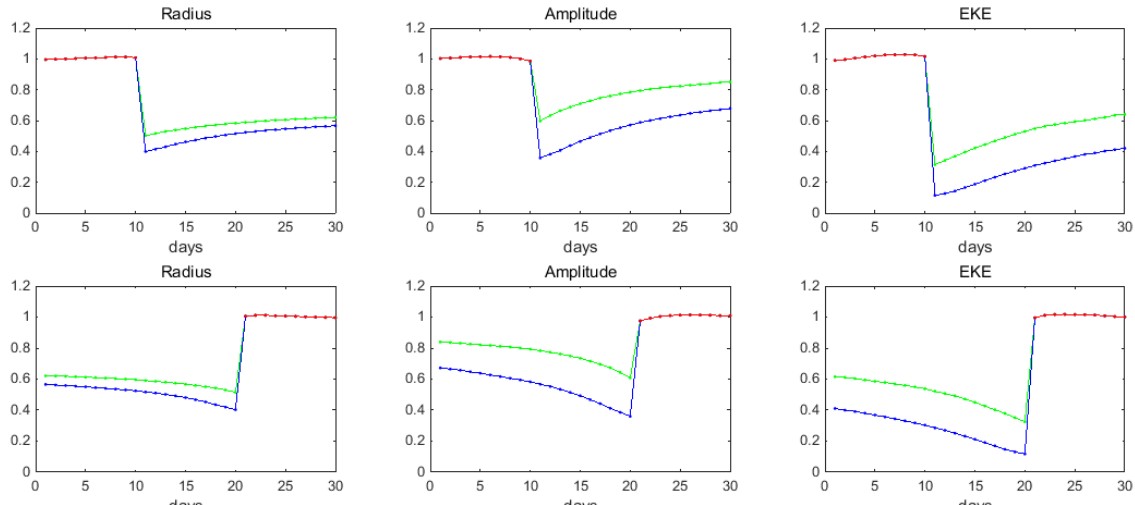


**Figure 10.** Average changes of eddy properties for all splitting (upper) and merging (lower) events. Properties of multicore eddies (red line) over 10 days and of single-core eddies (green and blue lines) over 20 days are presented. Properties of single-core eddies were normalized with respect to the multicore eddies.



## 5 Summary and conclusions

This study examined the global statistics of eddy splitting and merging based SLA data provided by AVISO from January 1993 to December 2015. Multicore structures were identified using a geometric closed-contour algorithm of SLA, which was improved in terms of certain technical details. Then, a two-dimensional anisotropic Gaussian surface fitting is used to confirm the multicore eddy structure rather than a misidentification of multiple eddies. Finally, a hybrid tracking strategy based on the eddy overlap ratio considering multicore and single-core eddies was used to confirm
splitting and merging events.

      Based on 23 years' satellite altimetry measurements, the census results showed 83,751 cyclonic and 83,406 anticyclonic multicore eddies with lifetimes of >6 days. About 95% of the multicore eddies had lifetimes of <30 days, and about 97% of the multicore eddies had propagation distances of <200 km. In comparison with single-core eddies, it was found that the lifetimes and propagation distances of multicore eddies tended to be much smaller because of their
unstable configuration. In fact it is an antinomy that on the one hand a multicore eddies only with few days (3 or 4 days) cannot be considered as a mesoscale or quasi-mesoscale process, while on the other hand a multicore eddy often has a short lifetime in the ocean because of its instability. Considering both aspects, only multicore eddy structures that existed for more than 6 days within a 10-day window were considered real eddy structures.

      The splitting and merging events were discerned from sequential time series of SLA maps. The process of eddy–
eddy interaction is firstly presented visually based on real sea surface height fields. Moreover, remote sensing SST data validated the eddy–eddy interaction. Generally, splitting or merging events can change the eddy scale, amplitude, and EKE substantially. In other words, the obvious variation (twice or more) of eddy properties for one full-lifetime eddy could correspond to a splitting or merging event. Merging events generally caused an increase of eddy properties, whereas splitting generally caused a decrease of eddy properties. Multicore eddies were found to tend to split into two
eddies with different intensities, with the larger one being on average almost twice the smaller one in terms of amplitude and EKE. Similarly, it was found that eddy merging tended not to involve the interaction of two equal-intensity eddies. Instead, a strong eddy tended to merge with a weaker one to form a larger multicore eddy in a process that appeared the reverse of the splitting process. In fact, multicore structures represent an intermediate stage in the process of eddy evolution, similar to the generation of multiple nuclei in a cell as a preparatory phase for cell division in biology. For
eddy splitting and for eddy merging, it is very important to identify the multicore eddies for studying eddy splitting and merging events and understanding the eddy–eddy interaction progress.

      Multicore eddies do not always correspond to splitting or merging. The hybrid tracking both considering multicore and single-core eddies globally detected 47,312 splitting events and 50,166 merging events for all 167,157 multicore eddies. Besides, about 14% of multicore eddies are transient eddy-like signals which do not match with single-core
eddies, and more than one quarter (28%) of multicore eddies neither split nor merge but are intermediate states of single-core eddy evolution.

      Geographic frequency statistics of splitting and merging events showed eddy–eddy interaction tended to occur in the Antarctic Circumpolar Current and western boundary currents, where typically about 5–7 events per $1°\times1°$ were observed. For the Northern Hemisphere, splitting and merging mainly occur in the Gulf Stream and its extension, the
Kuroshio Extension, the region of Subtropical Countercurrent in the Northwest-Pacific, the eastern Pacific coastal region,



and the region in the Gulf of Mexico. For the Southern Hemisphere, splitting and merging mainly occur in the Antarctic Circumpolar Current region and the region of 20 °–35 ° S. The areas of the Drake Passage, to the west of the Kerguelen Plateau, to the south of New Zealand, in the Gulf of Mexico, and along the axis of the Gulf Stream were found to have more splitting and merging events because of obvious topographic effects. Eddy splitting and merging do not always
occur most frequently in eddy-rich regions. Compared with the geographical distribution of single-core eddies, the Antarctic Circumpolar Current and Western Boundary Currents have higher frequencies of splitting and merging than mid-latitude regions (20 °–35 °) north and south which have more long-lifetime eddies. This implies that regions with strong current variation and obvious eddy–mean flow interaction where abundant eddies have shorter lifetimes, have more significant eddy–eddy interaction. Thus, when considering eddy splitting and merging, eddies in such regions could
have longer lifetimes than expected based on previous studies. Essentially, eddy splitting and merging are caused primarily by an unstable configuration of multicore structures due to obvious current– or eddy–topography interaction, strong current variation, and eddy–mean flow interaction.

It is interesting and instructive to compare the global ocean to the universe. The oceanic eddies are just like galaxies in the universe: both can spin around their cores, move in one direction, collide, split and merge, and finally disappear in
the background field. The variation in temperature and salinity fields caused by eddies in the ocean is similar with the space-time curvature caused by a galaxy in the universe. Haller and Beron-Vera (2013) even found coherent Lagrangian eddies can capture and swallow nearby passively floating debris that means eddies can be viewed as "black holes" in the ocean likely in cosmology.

Hybrid tracking considering single-core and multicore eddies for full-lifetime evolution is highly complex given
that some eddies might merge or split multiple times. Nevertheless, the description of full-lifetime eddy evolution needs to be addressed in future study. This work is very important for describing accurately the lifetime evolution of eddies in regions where substantial splitting and merging occur. The eddies in such regions are expected to have longer lifetimes. Limited by the satellite altimetry measurements, the surface eddy splitting and merging are analyzed in this study. For the subsurface information of eddy interaction we know nothing. This question is being addressed in ongoing research
from analysis of the altimeter data in combination with subsurface float observations and from the Parallel Ocean Program global ocean circulation model.

## Acknowledgements

This work was supported by the National Natural Science Foundation of China under contract No. 41576176, the National Key R&D Program of China under contract No. 2016YFC1401800, and in part by National Programme on
Global Change and Air-Sea Interaction. The altimeter products were produced by Ssalto/Duacs and distributed by Aviso, with support from Cnes (http://www.aviso.altimetry.fr/duacs/). The remote sensing SST data from AVHRR were provided by National Oceanic and Atmospheric Administration (NOAA).



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
