# Peer review of "Multicore structures and the splitting and merging of eddies in global oceans from satellite altimeter data"

_Ocean Science, 2018_

## Referee Comment (RC1) · Amores (Referee) · 18 Jan 2019

This manuscript deals with the problem of two eddies of the same polarity interacting between them (merging or splitting) from satellite altimetry observations. They apply a slightly modified eddy detection algorithm to identify eddy structures with one or more eddy cores and develop a new technique to study the phase of eddy merging or splitting. The results of the paper are interesting and it is well written. I recommend its publication after few considerations (they are not written by relevance order).

\* Although two cores are technically multiple cores, I would not use the word "multicore" since the authors only deal with structures of two cores ("... this study focused only on

the classical merging of two eddies into one and splitting of one eddy into two, ... "; line 413). This connects with the point number 3 of the section 2.2.1; there is no need to identify structures of 3 cores.

\* The authors say that "The census revealed that eddy splitting and merging do not always occur most frequently in eddy-rich regions,..." (line 17). When the map of eddy splitting and merging from Fig. 9 is compared with the distribution of eddy amplitude from Fig. 10 of Chelton et al. 2011, it can be seen the eddy splitting and merging takes place in regions with high eddy amplitude. I do not think that "stronger" eddies interact more than "weaker" eddies so it has to be related with the thresholds applied when identifying and tracking the eddies and with the capabilities of the satellite SLA maps.

\* The previous comment connects with the paper "Up to What Extent Can We Characterize Ocean Eddies Using Present-Day Gridded Altimetric Products?" by Amores et al. How do you think the results of that paper could influence the results of the present manuscript. In that paper it is showed that satellites tend to merge several small eddies into larger one, think that would make impossible to differentiate a real large eddy from a sum of several smaller eddies into larger one. Maybe you minimized this problem with the thresholds used and the different steps of the algorithms. However, I really think that it should be clarified in the paper: your results are from a very small fraction of the global ocean eddies, so they should not be taken as universal truth.

\* My next comment is out of the scope of this paper, but it could be interesting applying your algorithm to a model output and to satellite-like SLA maps extracted from the model (similar to the paper by Amores et al.) to see what part of the eddy interactions you are missing in your results.

\* What about the role of the eddy polarity in these processes. A Fig. 10 computed by polarity could be interesting.

\* Line 92: indicate the reference of the satellite product.

\* Line 95: What filtering processes were applied? Filtering window, type of filter, ...

\* Line 105 – 108: this could be moved to the part of data.

\*Line 113: ... subsequently transported.

\* The authors claim that their eddy tracking algorithm is parameter free. The step (6) of the eddy identification makes it whatever but parameter free.

\* Line 134: ... eddy core ...

\*Line 150: for an eddy are given.

\* Line 219: 2 altimeters were measuring during  $\sim$ 7 years (30%); 3 during  $\sim$ 9.5 years (41%); and 4.5 during  $\sim$ 6.5 years (29%). I do not know if these numbers support the statement "most times".

\* Line 467: "5-7 events". Are these events per year; events in the 23 years?

\* Figure 9: the colorscale is not adequate. Limiting the maximum allowed value will show the structures better. The units of the colorscale are missing. Change the text inside the panels to eddy merging and eddy splitting. It would be interesting to see a map of anticyclones merging (splitting) – cyclones merging (splitting).

\* Typos and similar: Line 2: ... an improved geometric... Line 33: ... local circulation of marginal seas ... Line 34: ... 2016), or the Mediterranean ... Line 482: ... the dramatically narrow ...

---

## Referee Comment (RC2) · Anonymous Referee #2 · 18 Feb 2019

The authors present a eddy detection algorithm which is able to identify multiple cores eddies structures which are mainly associated with splitting and merging events. If there is now a large number of eddy detection and tracking algorithm available, only a very few are able to identify merging and splitting events. For this reason this paper deserve attention. Such eddy-eddy interactions may indeed strongly impact the reconstruction of the eddy trajectory and its estimated lifetime if the tracking procedure does not account for them. However, few other methods were recently used to identify merging and splitting events from altimetry data sets and they should be properly referenced in the introduction. In a second stage, the authors provide statistical maps of eddy merging and splitting events on the global ocean for a 23 year-period. The

authors mention that splitting and merging events do not always occur most frequently in eddy-rich region. This is a very interesting aspect that should be deepened a little bit more.

Overall my recommendation is that the editor accept this article if the following remarks and suggestions are satisfactorily addressed.

1- Only few eddy detection and tracking algorithm are able to identify merging or splitting events. In order to describe the state of the art, the authors should refer to them explicitly in the introduction, especially in the fourth paragraph:

- Li, Q. Y., Sun, L., Liu, S. S., Xian, T., and Yan, Y. F.: A new mononuclear eddy identification method with simple splitting strategies, Remote Sensing Letters, 5(1), 65-72, 2014.

- Daisuke M., F. Araki, Y. Inoue, H. Sasaki, A New Approach to Ocean Eddy Detection, Tracking, and Event Visualization –Application to the Northwest Pacific Ocean, Procedia Computer Science,v 80,2016,1601-1611, https://doi.org/10.1016/j.procs.2016.05.491.

- Le Vu, B., A. Stegner, and T. Arsouze, 2018: Angular Momentum Eddy Detection and Tracking Algorithm (AMEDA) and its application to coastal eddy formation. J. Atmos. Oceanic Technol., 35, 739–762, https://doi.org/10.1175/JTECH-D-17-0010.1

- Laxenaire, R., Speich, S., Blanke, B., Chaigneau, A., Pegliasco, C., & Stegner, A. (2018). Anticyclonic eddies connecting the western boundaries of Indian and Atlantic Oceans. Journal of Geophysical Research: Oceans, 123. https://doi.org/10.1029/2018JC014270.

2-Page 9 , line 273 'Although a cyclonic eddy could theoretically merge directly with an anticyclonic eddy, the mixing process is too complex and the observation of such an event too difficult for the current research.' When opposite sign vortices get close to each other they tend to form a dipolar structure which propagate at a constant speed.

As far as I own no numerical simulations, laboratory experiments or remote sensing observation has shown that a cyclone and an anticyclonic could merge together ! This statement should be suppressed or precise references, showing such event, should be provided here.

3-Flow chart figure 2. It seems that a third arrow indicating the possbility of single eddy (>6days) with a transient double core structure should be present at the end of the flowchart between the splitting and the merging events. Such case should correspond to the column 5 of the table 1, if I'm not mistaken.

4- page 16 line 421: the authors do not provides here the number of (single-core) eddies with lifetimes > 6 days (numbers are given only for >30 days or >100 days) in order to compare with the number of multicore structures (> 6 days) mentioned just before.

5- page 17 lines 428-440 : the recent paper of Garreau et al. 2018 (https://doi.org 10.1029/2017JC013667 ) which depict the consecutive splitting and merging of the same anticyclone with its parent eddy could also be mentioned here.

6- Page 19, lines 485-500: the authors mention that merging and splitting events are not correlated to eddy-rich regions. It is indeed interesting to highlights specific areas where the ratio (eddy-eddy events)/(total nb of eddies) is higher than the statistical mean. However, the mentioned areas (Antarctic Circumpolar Current, the Gulf Stream and its extension, Kuroshio Extension, Agulhas Return Current, and Brazil–Malvinas Confluence Zone) seems to me 'eddy-rich regions'. A more quantitative analysis could be done here to provide such statistical ratio or the correlation between the eddy lifetimes and the splitting-merging events as suggested by the authors.

---

## Author Comment (AC2) · 17 Mar 2019

Reply to Referee comment 2

For a better reading experience, you can see the Supplement.

Thank you for your affirmation to our work and your valuable comments, it is very helpful for revising and improving our paper. We have studied comments carefully and have made correction which we hope meet with approval. The main corrections in the paper and the responds to the reviewer's comments are as following:

1 Only few eddy detection and tracking algorithm are able to identify merging or splitting

events. In order to describe the state of the art, the authors should refer to them explicitly in the introduction, especially in the fourth paragraph.

These references have been added and marked green in the third paragraph and the fourth paragraph.

2 Page 9 , line 273 'Although a cyclonic eddy could theoretically merge directly with an anticyclonic eddy, the mixing process is too complex and the observation of such an event too difficult for the current research.' When opposite sign vortices get close to each other they tend to form a dipolar structure which propagate at a constant speed. As far as I own no numerical simulations, laboratory experiments or remote sensing observation has shown that a cyclone and an anticyclonic could merge together! This statement should be suppressed or precise references, showing such event, should be provided here.

The sentence has been revised as "...a cyclonic eddy could theoretically interact directly with an anticyclonic eddy...", and the references have been provided. Amores et al. (2017) found that eddies in the global ocean would be surrounded by eddies of opposite polarity. Chang & Park (2015) investigated the temporal variation of the flow structure and consequent mixing process of a cyclonic mesoscale eddy as it collided with an anticyclonic eddy by analyzing the Hybrid Coordinate Ocean Model simulation for the Gulf Stream region. L'Hégaret et al. (2014) studied a collision of Mediterranean Water dipoles in the Gulf of Cadiz and found that the merger of two dipoles resulted in an anticyclone (a meddy) which drifted southeastward, coupled with the eastern cyclone.

Amores, A., Monserrat, S., Melnichenko, O., & Maximenko, N.: On the shape of sea level anomaly signal on periphery of mesoscale ocean eddies. Geophysical Research Letters, 44(13), 6926-6932, 2017. Chang, Y. S., & Park, Y. G.: Variation of flow properties during a collision event of two mesoscale eddies in the Gulf Stream region from numerical simulation. Ocean Science Journal, 50(3), 567-579, 2015. L'Hégaret, P.,

Carton, X., Ambar, I., Ménesguen, C., Hua, B. L., Chérubin, L., ... & Serra, N.: Evidence of Mediterranean water dipole collision in the Gulf of Cadiz. Journal of Geophysical Research: Oceans, 119(8), 5337-5359, 2014.

3 Flow chart figure 2. It seems that a third arrow indicating the possbility of single eddy (>6days) with a transient double core structure should be present at the end of the flowchart between the splitting and the merging events. Such case should correspond to the column 5 of the table 1, if I'm not mistaken.

It has been modified according to the comment.

4 Page 16 line 421: the authors do not provides here the number of (single-core) eddies with lifetimes > 6 days (numbers are given only for >30 days or >100 days) in order to compare with the number of multicore structures (> 6 days) mentioned just before.

Single-core eddies with lifetimes > 6 days are too short relative to the mesoscale phenomenon and do not represent a true oceanic eddy in the ocean. In another paper (The identification and census statistics of multicore eddies based on sea surface height data in global oceans, in press Acta Oceanologica Sinica), we compared the characteristics of multicore eddies and single-core eddies with lifetime > 30 days in global ocean. Only multicore structures that existed for more than 6 days within a 10-day window were considered real multicore eddy. Based on the multicore eddies, the eddy interaction is then studied by the matching of multicore eddies with single-core eddies. Because multicore eddies are an intermediate structure in the eddy evolution or interaction and their lifetime are expected to be shorter than single-core eddies, such an identification of multicore eddies with lifetimes > 6 days is adopted. If there is no single-core eddy with lifetime > 6 days before and after a multicore eddies, the multicore eddies will be considered as a transient eddy-like signatures.

5 Page 17 lines 428-440 : the recent paper of Garreau et al. 2018 (https://doi.org10.1029/2017JC013667 ) which depict the consecutive splitting and merging of the same anticyclone with its parent eddy could also be mentioned here.

[Figure]

The reference of Garreau et al. (2018) has been added and marked green in Section 4.

6 Page 19, lines 485-500: the authors mention that merging and splitting events are not correlated to eddy-rich regions. It is indeed interesting to highlights specific areas where the ratio (eddy-eddy events)/(total nb of eddies) is higher than the statistical mean. However, the mentioned areas (Antarctic Circumpolar Current, the Gulf Stream and its extension, Kuroshio Extension, Agulhas Return Current, and Brazil–Malvinas Confluence Zone) seems to me 'eddy-rich regions'. A more quantitative analysis could be done here to provide such statistical ratio or the correlation between the eddy lifetimes and the splitting-merging events as suggested by the authors.

Here, we compared splitting and merging events with frequency distribution of global eddies (Fig. 5 in Chelton et al. 2011) and found "Eddy splitting and merging events do not always occur most frequently in eddy-rich regions". Despite eddies are often generated and are high-amplitude in the Antarctic Circumpolar Current (ACC) and Western Boundary Currents (WBCs), the lifetime of eddies there is generally short and does not propagate too long. Therefore, the eddy frequency in ACC and WBCs may not be as obvious as eddies with long lifetime in the mid-latitude. So we came to this conclusion. Qualitative analysis shows that this phenomenon is easy to understand. The large-amplitude and high-strength eddies are more easily detached due to the instability of the flow in ACC and WBCs. Due to the significant interaction of eddy-current or eddy-topography, the instability of strong eddies is often caused, so the interaction events of eddy-eddy often occur. On the other hand, it is because of these interactions in ACC and WBCs that the lifetime of eddies here is shorter (although eddies with higher amplitude here) comparing with mid-latitude regions. Eddies in the mid-latitudes tend to have long lifetimes due to the ocean currents and eddy structures are stable with less (not zero) variation. As a result, census statistics for the numbers of eddy with long lifetime (Fig. 5 in Chelton et al. 2011) show the reduced number of eddies in ACC and WBCs and comparatively large numbers of eddies occurred in bands of

mid-latitude regions of 20°–35° north and south. Hybrid tracking considering single-core and multicore eddies for full-lifetime evolution is highly complex given that some eddies might merge or split multiple times. Nevertheless, the description of full-lifetime eddy evolution needs to be addressed, and a comparison with the lifetime of traditional single-core eddy evolution without considering eddy-eddy interaction will be carried out in future study. And we are also ready to consider the relationship between eddy properties (e.g., eddy amplitude or eddy intensity) and the eddy-eddy interaction, which will be the focus of future research. This article focuses on abundant multicore structures and the eddy-eddy interaction (splitting and merging) in global oceans

We tried our best to improve the manuscript and made many changes in the manuscript. The other important changes are marked in green in revised paper. We did not list all changes which not influence the content and framework of the paper, especially for the changes of grammar and written expression. We appreciate for Editors/Reviewers' warm work earnestly, and hope that the correction will meet with approval.

Sincerely, Authors

Please also note the supplement to this comment:
https://www.ocean-sci-discuss.net/os-2018-96/os-2018-96-AC2-supplement.pdf

———————————————

---

## Author Comment (AC1)

**Reply to Referee comment 1**

Dear Amores,

Thank you for your comments concerning our manuscript entitled "Multicore structures and the splitting and merging of eddies in global oceans from satellite altimeter data". Those comments are all valuable and very helpful for revising and improving our paper. We have studied comments carefully and have made correction which we hope meet with approval. The main corrections in the paper and the responds to your comments are as following:

1   *Although two cores are technically multiple cores, I would not use the word "multicore" since the authors only deal with structures of two cores ("...this study focused only on the classical merging of two eddies into one and splitting of one eddy into two, ... "; line 413). This connects with the point number 3 of the section 2.2.1; there is no need to identify structures of 3 cores.*

In another paper (*The identification and census statistics of multicore eddies based on sea surface height data in global oceans*, in press Acta Oceanologica Sinica), we produced a statistical analysis of multicore eddy structures based on 23 years' altimetry data in global oceans. Census statistics of multicore eddies present about 97 % of all eddies are dual-core structures and about 3% are triple-core structures. The significant difference between the numbers of dual and triple-core eddies shows that the former are more common and easily detected in the ocean. Even so, the triple-core eddies cannot be ignored in the global ocean. Although these triple-core structures may not represent really oceanic eddies (possibly a confusion with some smaller eddies). Identification of these eddies is very helpful to track eddy trajectories in the ocean based on the current resolution of sea surface height field from multi-altimeter products (As shown in the following figure). The global statistic shows that there are more than 28% of multicore eddies identified as part of the evolution of single-core eddies, of which are some triple-core eddies. So when possible, we have also identified this small number of triple-core structures (there is no technical problem, maybe the real state of these eddies on the ocean needs further verification.).

The identification of multi-core eddies is mainly to provide a technical support as an analytical basis for interactions between multiple eddies (two or more). Therefore, it is considered here that multi-core eddies is still used.

[Figure]

A dual-core eddy evolves as a part of a single-core eddies. These multicore eddies represent intermediate states of single-core eddy evolution. If we do not identify the multicore eddy, the eddy with two cores that are very close together will not recognized in the traditional procedure of eddy detection. In that case, the single-core eddy before and after multicore eddy will be likely to identified as two independent eddies. This is an example of a dual-core eddy, some of triple-core eddies may also correspond to this situation.

2    *The authors say that "The census revealed that eddy splitting and merging do not always occur most frequently in eddy-rich regions,..." (line 17). When the map of eddy splitting and merging from Fig. 9 is compared with the distribution of eddy amplitude from Fig. 10 of Chelton et al. 2011, it can be seen the eddy splitting and merging takes place in regions with high eddy amplitude. I do not think that "stronger" eddies interact more than "weaker" eddies so it has to be related with the thresholds applied when identifying and tracking the eddies and with the capabilities of the satellite SLA maps.*

In this study, the minimum amplitude of an eddy was increased from the original 1 cm used by Chelton et al. (2011) to 3 cm. The reason for this change was that the accuracy of measuring heights using Jason series altimeters (including Topex/Poseidon and Jason-1/2/3), which currently have optimal performance for observing ocean dynamics, is only about 2 cm in the open sea (Dufau et al., 2016). Therefore, even though the AVISO gridded SLA products represent the merging of data from different altimeters, it is difficult to claim that ocean signals under a variance of 2 cm could be captured precisely in the SLA fields, especially for the gaps in altimeter tracks that are interpolated from other observation points.

We checked the mapping error of AVISO gridded products. It mainly traduces errors induced by the constellation sampling capability and consistency with the spatial/temporal scales considered, as described in Le Traon et al. (1998) or Ducet et al. (2000). The mapping error is less than 2 cm in the open ocean and just may be greater than 3cm in regions of highly unstable currents, e.g., Antarctic Circumpolar Current, some Western Boundary Currents (the figure below or Dufau et al., 2016). That is why **the amplitude of an eddy greater than 3cm be adopted in the eddy identification rather than 1 cm** like in Chelton et al. (2011). It is an effective way to avoid artefacts from AVISO and large, ameba-like eddy structure in the oceans.

Of course, sea surface height fields with higher resolution can show more mesoscale (sub-mesoscale) ocean dynamics (Amores et al., 2018). If we can further improve the spatial resolution, we will see more small-scale ocean structures. However, based on the current mainstream SSH products with 0.25° spatial resolution, we can only avoid the eddy misidentification as much as possible by increasing the threshold of eddy amplitude, and by identifying the eddy with a radius scale above 50 km.

[Figure]

The mapping error of AVISO SLA gridded products on 10 March, 2015. (unit: cm) (From AVISO)

On the other hand, our result is not to say that "stronger" eddies interact more than "weaker" eddies (although it looks like that, globally). This result should be understood in this way. The large-amplitude and high-strength eddies are more easily detached due to the instability of the flow in the Antarctic Circumpolar Current (ACC) and Western Boundary Currents (WBCs). Due to the significant interaction of eddy-current or eddy-topography, the instability of strong eddies is often caused, so the interaction events of eddy-eddy often occur. In turn, it is because of these interactions in ACC and WBCs that the lifetime of eddies here is shorter (although eddies with higher amplitude here) comparing with mid-latitude regions.

Of course, if we focus on some local regions, such as the Gulf Stream and its extension, weaker eddies may be more likely to interact with each other than stronger eddies (theoretically because the former is more unstable). A statistical study of this situation will be carried out in future. However, globally, the high-amplitude eddy is in ACC and WBCs, where there are more unstable factors than the mid-latitude region. Although the eddy intensity is stronger, it is more likely to eddy-eddy interaction or eddy deformation due to unstable factors (e.g., westward eddy interact with eastward current in ACC), so the lifetime of eddies is much shorter (the result is very clear in the Fig4 and Fig.10 in Chelton et al. 2011) in ACC and WBCs.

The corresponding descriptions are added in the Line 507-513.

Chelton, D. B., Schlax, M. G., & Samelson, R. M. (2011). Global observations of nonlinear mesoscale eddies. Progress in Oceanography, 91(2), 167-216.
Ducet, N., Le Traon, P. Y., & Reverdin, G. (2000). Global high‐resolution mapping of ocean circulation from TOPEX/Poseidon and ERS‐1 and‐2. Journal of Geophysical Research: Oceans, 105(C8), 19477-19498.
Dufau, C., Orsztynowicz, M., Dibarboure, G., Morrow, R., & Le Traon, P. Y. (2016). Mesoscale resolution capability of altimetry: Present and future. Journal of Geophysical Research: Oceans, 121(7), 4910-4927.
Le Traon, P. Y., Nadal, F., & Ducet, N. (1998). An improved mapping method of multisatellite altimeter data. Journal of atmospheric and oceanic technology, 15(2), 522-534.
Amores, A., Jordà G., Arsouze, T., & Le Sommer, J. (2018). Up to What Extent Can We Characterize Ocean Eddies Using Present‐Day Gridded Altimetric Products?. Journal of Geophysical Research: Oceans, 123(10), 7220-7236.

3    *The previous comment connects with the paper "Up to What Extent Can We Characterize Ocean Eddies Using PresentâˇARˇ Day Gridded Altimetric Products?" by Amores et al. How do you think the results of that paper could influence the results of the present manuscript. In that paper it is showed that satellites tend to merge*

*several small eddies into larger one, think that would make impossible to differentiate a real large eddy from a sum of several smaller eddies into larger one. Maybe you minimized this problem with the thresholds used and the different steps of the algorithms. However, I really think that it should be clarified in the paper: your results are from a very small fraction of the global ocean eddies, so they should not be taken as universal truth.*

Based on sea surface height products with 0.25 ° resolution, the automatic eddy detection and tracking algorithm would potentially be unable to characterize the mesoscale variability (Amores et al. 2018). The main reason is that the spatial resolution of the gridded products is not enough to capture the small-scale eddies that are the most abundant (But then, if we can further improve the spatial resolution in numerical simulations, we will see more small-scale ocean structures). Also, the unresolved structures are aliased into larger structures in the gridded products, so those products show an unrealistic number of large eddies with overestimated amplitudes.

However, in our paper we present a hybrid tracking strategy based on the eddy overlap ratio considering multicore and single-core eddies and a statistical method of eddy-eddy interaction. And based on the current mainstream SSH products with 0.25 ° spatial resolution, the global statistics of eddy splitting and merging are examined, which has a kind of enlightening significance. If there are higher spatial resolution and more accurate products from numerical simulations or assimilated results, a further detailed map of the eddy-eddy interaction in global oceans may be obtained using this method.

To avoid artefacts from AVISO and large, ameba-like eddy structure in the oceans, eddies with large spatial scale and much larger amplitude are identified in our paper. And two examples of verification using SST data are given. Under the current SSH product, we can't completely avoid this kind of misidentification.

The corresponding descriptions are added in the last paragraph in the conclusions.

4    *My next comment is out of the scope of this paper, but it could be interesting applying your algorithm to a model output and to satellite-like SLA maps extracted from the model (similar to the paper by Amores et al.) to see what part of the eddy interactions you are missing in your results.*

In the future, we can try to do this.

5    *What about the role of the eddy polarity in these processes. A Fig. 10 computed by polarity could be interesting.*

The cyclonic eddy and the anticyclonic eddy exhibit similar variations of eddy properties for eddy splitting and merging events. Merging events generally caused an increase (twice or more) of eddy properties, whereas splitting generally caused a decrease (halved) of eddy properties. Here we do not separately compute the variations of eddy properties for different eddy polarity.

6    *Line 92: indicate the reference of the satellite product.*

The reference of the satellite product is added in Section 2.1.

7    *Line 95: What filtering processes were applied? Filtering window, type of filter, ...*

The sentence should be "Filtering processes were used to remove residual noise and small-scale signals in the procedure of multi-altimetry data by the AVISO".

Residual noise and small scale signals are then removed by filtering the data using a Lanczos filter. As data are filtered from small scales, a sub-sampling is finally applied. The filtering and sub-sampling is adapted to each region and product as a function of the characteristics of the area and of the assimilation needs. Details were presented in Dufau et al., 2013 and AVISO, 2015.

AVISO. (2015). SSALTO/DUACS User Handbook: (M)SLA and (M)ADT Near-Real Time and Delayed Time Products. CLS-DOS-NT-06-034 - Issue 4.4

Dufau, C., Labroue, S., Dibarboure, G., Faugère, Y., Pujol, I., Renaudie, C., & Picot, N. (2013, October). Reducing altimetry small-scale errors to access (sub) mesoscale dynamics. In Ocean Surface Topography Science Team Meeting (Vol. 811).

8    *Line 105 – 108: this could be moved to the part of data.*
     This paragraph has been moved to Section 2.1.

9    *Line 113:… subsequently transported.*
     It has been modified according to the comment.

10   *The authors claim that their eddy tracking algorithm is parameter free. The step (6) of the eddy identification makes it whatever but parameter free.*
     We mistakenly adopted the method name "threshold-free closed-contour algorithm" from Chelton et al. (2011) because they detected the eddy signals with amplitude exceeded 1 cm. Here, we revised it as "threshold closed-contour algorithm" in our paper.

11   *Line 134: … eddy core …*
     It has been modified according to the comment.

12   *Line 150: for an eddy are given.*
     It has been modified according to the comment.

13   *Line 219: 2 altimeters were measuring during ~7 years (30%); 3 during ~9.5 years (41%); and 4.5 during ~6.5 years (29%). I do not know if these numbers support the statement "most times".*
     We have changed "most times" to "after the year 2000".

14   *Line 467: "5-7 events". Are these events per year; events in the 23 years?*
     It is over the totally 23-year period.

     Here we used a more rigorous tracking method to study eddy splitting and merging. That is, a multicore eddy needs to appear in the evolution of a single-core eddy and then splits into two single-core eddies, which dynamic process is just considered to be a splitting event (an opposite process for merging event). This requirement is very demanding, so the census statistics of splitting and merging events may be much less than those in real oceans. Generally, splitting or merging events can change the eddy scale, amplitude, and EKE substantially. In another words, the obvious variation (twice or more) of eddy properties for one full-lifetime eddy could correspond to a splitting or merging event. If the eddy is considered to merging or splitting when the eddy property changes more than 1 times during the entire lifetime of eddy evolution, then the number of such eddy merging and splitting event in the broad sense will increase much more than the original analysis. At the moment we are doing something like this.

15   Figure 9: the colorscale is not adequate. Limiting the maximum allowed value will show the structures better. The units of the colorscale are missing. Change the text inside the panels to eddy merging and eddy splitting. It would be interesting to see a map of anticyclones merging (splitting) – cyclones merging (splitting).
     The colorscale has been readjusted to show the structures better.

     Geographic frequency statistics of the splitting and merging events for cyclones and anticyclones are shown in the figure below. The cyclones and the anticyclones exhibit similar distribution characteristics, eddy splitting and merging tend to occur in the ACC and WBCs which is similar to the Figure 9 in our paper. The map of anticyclones merging (splitting) – cyclones merging (splitting) shows a lot of irregular mosaic features, which is not given in the original paper. The difference of eddy merging or splitting for polarity may be

obvious in some local regions if sub-mesoscale (or smaller) eddies was considered based on high-resolution SSH products (Amores et al. 2018 shown vast majority of the eddy field in AVISO SSH product is missed because the available observations do not have enough resolution to resolve the smaller eddies).

Here, in our paper based on the current mainstream SSH products with 0.25 ° spatial resolution, we focus on multicore eddies that experienced splitting or merging, geographic frequency statistics of eddy splitting and merging, and variations of eddy properties. If there are higher spatial resolution and more accurate products from numerical simulations or assimilated results, a further detailed map of the eddy-eddy interaction (also differences between anticyclones and cyclones) in global oceans may be obtained using this method.

[Figure]

Census statistics for cyclones(left)/anticyclones(right) splitting and merging events (number of events) for each 1 ° × 1 ° region over the 23-year period (January 1993 to December 2015).

16  *Typos and similar: Line 2: ... an improved geometric:... Line 33: ... local circulation of marginal seas ... Line 34: ... 2016), or the Mediterranean...Line 482: ... the dramatically narrow ...*

It has been modified according to the comment.

We tried our best to improve the manuscript and made many changes in the manuscript. The other important changes are marked in green in revised paper. We did not list all changes which not influence the content and framework of the paper, especially for the changes of grammar and written expression. We appreciate for Editors/Reviewers' warm work earnestly, and hope that the correction will meet with approval.

Sincerely,

Authors